# From Policies to Practices: Factors Related to the Use of Inclusive Practices in Portugal

Joana Cruz [1,*,†] , Helena Azevedo [2,†] , Marisa Carvalho [3] and Helena Fonseca [4]

1 Psychology of Development Research Centre, Institute of Psychology and Education Sciences, Lusíada University, 4100-348 Oporto, Portugal
2 Research Unit in Psychology and Human Development, Department of Social and Behavioral Sciences, University of Maia, 4475-690 Maia, Portugal; hazevedo@umaia.pt
3 Research Centre for Human Development, Faculty of Education and Psychology, Universidade Católica Portuguesa, 4169-005 Oporto, Portugal; mscarvalho@ucp.pt
4 Inspectorate for Education and Science, 1350-346 Lisbon, Portugal; helena.fonseca@igec.mec.pt
* Correspondence: joanacruz@por.ulusiada.pt
† These authors contributed equally to this work.

**Abstract:** Inclusion is considered a foundation for quality education, and teachers' inclusive practices are essential for success in mainstream classrooms. Portugal has been making progressive improvements in its policies for inclusive education, although there is little consistency in school practices within or between schools. Moreover, data identifying the personal and career variables relevant to teachers' inclusive practices in Portugal are scarce. Therefore, the purpose of this study was to determine the relationship between teachers' inclusive practices and personal and career-based characteristics, including gender, level of teaching, years of experience, roles performed at school, and perception of inclusive resources. The participants were 924 teachers who worked in private and public schools in Portugal. Regression analysis showed that perceived inclusive resources, level of teaching, and gender predicted variance in inclusive practices. Mean difference analyses revealed that teachers at the lower levels of teaching, females, and teachers reporting more inclusive resources had the highest scores for inclusive practices. These findings are discussed in terms of their practical relevance for inclusive school systems.

**Keywords:** inclusive practices; inclusive resources; level of teaching; gender; inclusive education

## 1. Introduction

Education systems worldwide face challenges directed toward including all children in schools. Fostering a more inclusive and equitable education is a central priority for politicians, researchers, and educational agents [1,2]. In the Education 2030 Agenda [3], inclusion and equity are set as the foundations for quality education, as expressed in its fourth Sustainable Development Goal (SGD4)—"Ensuring Inclusive and Equitable Quality Education and Promoting Lifelong Learning Opportunities for All". Education is considered a fundamental right and an enabling right, and no one should be left behind. Thus, it is necessary that countries worldwide create and implement policies and practices that contribute to schools becoming more inclusive. This specifically calls for addressing inequalities in access, participation, learning processes, and outcomes [4].

Although the relevance of promoting inclusive education is generally recognized, the meaning of this concept remains an unclear and widely debated issue [5]. While narrow definitions are more limited in scope and tend to focus on specific groups of students in vulnerable situations, a broader definition gives value to diversity and emphasizes this concept as an approach to support all students [6]. Consistent with this latter view, inclusion is considered to be a process that involves identifying and removing barriers to access, learning, and achievement for all students [1], as well as the ideal outcome of such

practices [7]. As either a process or an outcome, inclusive education can take many forms, and the practices teachers should use in classrooms are diverse [8,9].

*1.1. Literature Review*

Developing an inclusive education requires teaching practices that focus on creating environments and opportunities that encourage teaching and learning processes while also considering student diversity [1,4,10]. Inclusive pedagogy is a critical factor for high-quality inclusive education [11] and encompasses practices that overcome barriers to the participation and learning of students [12]. As Woodcock and colleagues [13] (2022, p. 3) argued: "these inclusive practices include the capacity to modify and differentiate instruction to accommodate student diversity and to set personalized goals which are appropriate to students' profiles".

The European Agency for Development in Special Needs Education [14] has recognized the complexity and diversity of practices related to inclusive teaching. Indeed, in addition to the complex definition, it is difficult to condense all practices into a single set of practices that can be applied in different contexts. Although inclusion practices can be diverse and look different in different contexts [15], research has identified several features related to inclusive practices. For instance, findings from a systematic review [4] indicated that collaboration and co-teaching, grouping, modification (of assessment, content, extent, instruction, learning environment, material, process, product, and time frame), individual motivation and feedback, and personal support of students were all characteristics of inclusive practices. Corroborating this result, a scoping review by Finkelstein et al. [5] highlighted collaboration and teamwork, instructional practices, organizational practices, social/emotional/behavioural practices, and determinations of progress as domains related to inclusive teaching practices. Other research has found that inclusive practices include the celebration of diversity, teaching planning, the taking into account of all students, education process, a varied methodology, formation of flexible and heterogeneous groups, the organization of times and spaces, support, and evaluation and transit between education stages [16,17].

Recent research into the perceptions of teachers and students has shown that teachers use inclusive teaching practices [11,18]. Moreover, the literature underlines the important role of teachers in the success of inclusive education [15], requiring that teachers accept responsibility for creating schools where all students can learn and feel a sense of belonging [19]. Several factors related to teachers' personal and career characteristics, as well as contextual factors, have been found to influence teaching practices [11]. Recent research has focused on teachers' attitudes toward inclusion and the influence of factors such as self-efficacy and beliefs about inclusive education. However, research into the factors influencing teachers' practices is scarcer [15]. In the current study, we focused on variables related to teachers' personal and career characteristics (gender, teachers' roles, level of teaching, and years of experience) and inclusive resources (material, technological, and human) and explored their relationships with inclusive practices.

Research into the use of inclusive practices at different school levels showed that primary education teachers used more inclusive teaching practices than did secondary education teachers [11,20]. One possibility is that this result reflects the organization of the education system, with primary schools providing one teacher per class versus one teacher per subject at the secondary level, because this means it is likely the former use more individualized teaching practices [11].

It is relevant to consider gender as a potential variable because, to the best of our knowledge, studies evaluating the effect of gender on inclusive practices are scarce. Within the few existing studies, Gebhardt et al. [20] assessed the influence of gender on classroom-inclusive practices such as teamwork and collaboration between general and special education teachers. However, this study found no effect associated with gender. Other research has analysed the influence of gender on other variables, such as attitudes and self-efficacy towards inclusive education, which evidence suggests are positively associated with the

use of inclusive practices [18,21]. However, research into gender and inclusive attitudes has been inconclusive. While some studies found that male and female teachers were similar in their inclusive attitudes [22–24], others reported more positive attitudes in female teachers [25,26], and yet others have identified more positive attitudes in male teachers [27]. Studies into gender and self-efficacy have also been inconclusive. For example, while some studies concluded that female teachers had higher levels of self-efficacy [28,29], others found higher levels of self-efficacy in male teachers [30]. Other studies did not identify a significant difference between male and female teachers in terms of their self-efficacy in teaching in inclusive classrooms [31].

Research evidence has also been inconsistent regarding the relationship between years of experience in teaching and the influence on inclusive teaching practices. While [11] reported no significant differences in inclusive practices between expert and novice teachers, a study by Schwab et al. [18] found, in contrast, that years of experience was a predictor of teaching practices. In this latter study, Schwab et al. [18] collected the perceptions of secondary school students and their teachers about inclusive teaching practices in the classroom for different subjects. The results showed that teachers with more extended experience tended to use more personalization. Krischler et al. [32] obtained similar results, showing that expert teachers considered inclusive education as a way to reach all students and perceive themselves as adopting more inclusive practices than do novice teachers.

*1.2. The Portuguese Context*

Recent years have seen Portugal improve its policies and practices toward inclusive education [33]. In 2018, the Portuguese government enacted a law, underpinned by a whole-school approach, which enhanced the inclusion of all students regardless of their personal or social conditions (Decree-Law 54/2018). In this way, inclusive education was assumed to be a process used to respond to the diversity of students' needs by increasing the participation of all students in learning and school life. Inclusive education is defined in a broader perspective as "the right of all children and pupils to access and participate, fully and effectively, in the same educational contexts" (Decree-Law 54/2018, Art. 3c). This legal framework is acknowledged internationally as a progressive law [10] that provides support for all students at mainstream schools without labelling them. Categorizing students as a way to determine who receives support is abandoned in favour of addressing and overcoming learning and inclusion barriers [33]. Nevertheless, this law advocates a multilevel approach to accessing the curriculum, recognizing that students have different support needs. Thus, the focus of the law is a pedagogical model based on the idea that all students have learning potential as long as they have adequate support [34]. This model implies a shift in focus from 'what is wrong with the child?' to 'what does the child need to support their learning?' [19]. This conceptual change highlights teachers' practices as being central to supporting all students' needs, overcoming learning and inclusion barriers, and ensuring inclusive learning environments.

Although the Portuguese education system endeavours to develop inclusive schools through policies founded on inclusive principles, translating these policies into school practices is more complex and challenging [34]. Policies are relevant for supporting conceptualizations and practices, but more is needed to ensure inclusive practices. This requires changes in thinking and practice at various levels of the education system, such as classroom teachers and other stakeholders and education leaders, as well as the wider community [35]. There appears to be little consistency in Portuguese inclusive practices within and between schools, meaning that policies have been translated into different practices as a function of context [34]. This is further complicated by the fact there is no single model for implementing inclusion at school and structuring teaching in the classroom to meet the individual needs of students [18].

The current literature suggests there is a need to further study the influence of teachers' roles in inclusive education [36]. Moreover, the recent shift in the Portuguese legal framework on inclusive education brought new opportunities and challenges to schools and the

management of teachers' roles, namely, within the participation of teachers in multidisciplinary teams and the attributions and functions of mainstream and special education teachers [37]. On the one hand, shared responsibility for the learning and development of all students requires an expansion of the roles and duties of special education teachers, from a more traditional remedial role to a consultative and supportive role, through a collaborative relationship with mainstream teachers in planning and preparing instruction for all students. In this way, special education teachers can be perceived as a resource for the whole school [38]. On the other hand, implementing inclusive practices while acknowledging classroom diversity and assuming a proactive approach to identifying and removing learning barriers is a requirement for mainstream teachers. In this way, inclusive education demands shared responsibility and a collaborative approach. In the context of an assessment of the challenges facing inclusive education in Portugal, Alves and colleagues [34] highlighted teachers' resistance to assuming the new roles required by the new legal framework. Teachers highlighted the lack of resources and training as challenges in their practice. Consequently, given the recent law of inclusive education, it is essential to explore the influence of teacher roles on inclusive practices in Portugal. Moliner et al. [39], based on an implementation of an index for inclusion for Spanish teachers in an inclusive classroom in secondary education, concluded that special education teachers are more sensitive to diversity and more aware of inclusive pedagogical practices than are general teachers.

Resources are widely identified by teachers as a relevant variable for inclusive education. Following Decree-Law 54/2018, an analysis of challenges to inclusive education in Portugal identified the lack of human resources (teachers and other professionals) and insufficient professional development opportunities related to inclusion as barriers to inclusive classroom practices [40]. In addition, Alves et al. [34] highlighted that resources, such as specialized human resources, were perceived by teachers as being essential for the implementation of the recent legal framework. Furthermore, teachers seemed to perceive that additional support staff were necessary to enhance the learning and participation of all students, particularly for students who experience difficulties and need additional support [9,13,34].

Success in implementing inclusive education policies is largely dependent on teachers' practices. This considered, we present an exploratory study of the inclusive practices of teachers from Portuguese public and private schools. This study aimed to investigate the relationship between teachers' inclusive practices and their personal and career characteristics and to assess how inclusive practices differ as a function of gender and career characteristics. To this end, we hypothesized that gender, perceived inclusive resources, number of years of experience, roles performed at school, and level of teaching would predict inclusive practices. We also hypothesized that subgroups of teachers differing in personal and career characteristics (e.g., male vs. female) would differ in their levels of inclusive practices.

## 2. Materials and Methods

### 2.1. Participants

The sample comprised 924 teachers who worked in private and public schools in Portugal (751 female, $M_{age}$ = 49.70, $SD$ = 7.47). Most participants had substantial professional experience in teaching (i.e., over 21 years of experience), were older than 51 years old, and taught at a public school. Women were overrepresented in the sample (81.3%), consistent with the unequal distribution of gender in the education system in Portugal [41,42]. In Portugal, early childhood education occurs from ages 3 to 6. After this, the first level of basic education includes students from ages 6 to 10, the second level of basic education includes students from ages 10 to 12, and the third level of basic education includes students from ages 12 to 15. Finally, secondary education includes students from ages 15 to 18. The participants taught at all these different levels of education or special education, although the largest proportion taught at the third level of basic education. A small proportion of the

participants (8.9%) taught at more than one education level. Most participants did not have additional roles in addition to teaching, and most did not have postgraduate qualifications (see Table 1).

**Table 1.** Participants' demographic information.

| Variables | N | % |
|:---:|:---:|:---:|
| Age | | |
| Under 30 years | 11 | 1.2 |
| 31–40 years | 95 | 10.3 |
| 41–50 years | 378 | 40.9 |
| Above 51 years | 440 | 47.6 |
| Gender | | |
| Male | 173 | 18.7 |
| Female | 751 | 81.3 |
| Qualifications | | |
| Undergraduate | 610 | 66.0 |
| Postgraduate | 314 | 34.0 |
| Number of years of experience | | |
| Under 10 years | 53 | 5.7 |
| 11–20 years | 235 | 25.4 |
| 21–30 years | 394 | 42.6 |
| 31–40 years | 228 | 24.7 |
| Above 41 years | 12 | 1.3 |
| Type of school | | |
| Public | 885 | 95.8 |
| Private | 39 | 4.2 |
| Level of teaching | | |
| Early childhood | 80 | 8.7 |
| 1st level | 150 | 16.2 |
| 2nd level | 124 | 13.4 |
| 3rd level | 235 | 25.4 |
| Secondary | 168 | 18.2 |
| More than one level | 82 | 8.9 |
| Special education | 85 | 9.2 |
| Roles | | |
| General council | 13 | 1.4 |
| Top leadership | 79 | 8.5 |
| Intermediate leadership | 150 | 16.2 |
| Class coordinator | 150 | 16.2 |
| Other coordination roles | 54 | 5.8 |
| Without additional roles | 478 | 51.7 |

[1] N = 924.

### 2.2. Measures

The 24 items of the Resources and Practices for Inclusion (RPI) [43] questionnaire capture two dimensions: inclusive resources and inclusive practices. All items had a 5-point Likert-scale response format (1—completely disagree to 5—completely agree), with higher scores indicating greater inclusivity. The Inclusive Resources subscale comprised nine items related to human (e.g., "The staff at the school includes enough specialists/auxiliary workers to attend to its student diversity"), technical (e.g., "The school's equipment and furniture are adapted to students' needs") and technological (e.g., "The computer rooms are equipped with enough computers for the numbers of students") resources used to promote learning.

The Inclusive Practices subscale comprised 15 items related to beliefs (e.g., "Student diversity enriches the education process") and behaviours (e.g., "I have extra activities for students who finish tasks early") that promote learning. Both subscales have been shown to have good psychometric properties, with Cronbach alpha values of 0.815 (Inclusive Resources) and 0.902 (Inclusive Practices). The Cronbach alpha values for the

current sample were 0.821 for the inclusive resources dimension and 0.902 for the inclusive practices dimension.

The sociodemographic questionnaire included questions on participants' characteristics (gender, age, and qualifications) and career experience (type of school, number of years of experience, roles performed at school, and level of teaching).

### 2.3. Procedures

The only precondition for participation in the study was being a teacher working in a Portuguese public or private school. We presented all relevant information for participants to give their informed consent, following APA ethical standards, and participants were only able to complete the questionnaire after giving their consent. Included in this information, participants were provided with the email of the principal investigator so they could ask questions before, during, and/or after the study. Participants were asked to complete an online self-report questionnaire, which was estimated to take 10 to 15 min. Participants were assured of confidentiality and informed that their participation was voluntary. The aims of data collection were summarized.

Participants were presented with the online questionnaire through mailing lists delivered through school directors. Opening the questionnaire link provided participants with a complete description of the objectives, institutional framework, length, and confidentiality issues. If participants chose to complete the questionnaire, they were presented with an online consent form. The questionnaire was available online between March and June 2019. Volunteers did not receive any compensation for their participation. The study did not request information that could allow participants to be identified.

### 2.4. Data Analysis

Data were analysed using an IBM SPSS v26. Analyses of variance (ANOVAs), independent samples *t*-tests, and multiple linear regression analysis were used to test the study's hypotheses.

First, to characterize the sociodemographic and career-based characteristics of the participants, we performed a descriptive analysis. Next, we used multiple linear regression to test the extent to which gender, inclusive resources, years of experience, roles performed at school, and level of teaching predicted variance in inclusive practices. For this linear model, we re-coded categorical variables with more than two categories into dichotomous variables. Thus, the six school roles were recoded into class coordinator vs. other roles due to the proximity to students' criteria. The seven categories for levels of teaching were recoded into initial schooling (including early childhood and 1st level) vs. other levels of schooling.

As a preliminary step, we performed a correlational analysis to evaluate the relationships between the outcome (Inclusive Practices) and predictor variables, with the intention that variables not significantly correlated to the outcome variable were not included in the regression model. As seen in Table 2, years of teaching experience was uncorrelated with Inclusive Practices ($r = -0.028$, $p > 0.05$).

One regression model, including gender, inclusive resources, roles performed at school, and level of teaching as predictors of inclusive practices, was tested. A stepwise approach was chosen due to the study's exploratory nature [44]. All assumptions for regression analyses were tested. An analysis of standard residuals was performed, and this indicated that there were no outliers. Analyses of standard residuals confirmed that the data contained no outliers. Tests to see if the data met the assumption of collinearity indicated that multicollinearity was not a concern (tolerance values from 0.959 to 1.00; VIF ranged from 1.00 to 1.04). The data also met the assumption of independent errors (Durbin–Watson = 1.92). The histograms of standardized residuals indicated that the data contained approximately normally distributed errors, confirmed with the normal P-P plots of standardized residuals, and the scatterplots of standardized residuals showed that the data met the assumptions of homogeneity of variance and linearity.

**Table 2.** Correlation matrix depicting correlations between the outcome and predictor variables.

| Variable | 1 | 2 | 3 | 4 | 5 | 6 |
|---|---|---|---|---|---|---|
| 1. Inclusive Practices | - | 0.245 *** | −0.136 *** | 0.067 * | 0.124 *** | −0.028 |
| 2. Inclusive Resources | | - | 0.142 *** | 0.016 | 0.029 | −0.087 ** |
| 3. Level of Teaching [a] | | | - | −0.216 *** | −0.141 *** | −0.108 ** |
| 4. Roles [b] | | | | - | 0.015 | 0.041 |
| 5. Gender [c] | | | | | - | 0.019 |
| 6. Years of Experience | | | | | | - |

N = 922. *** $p < 0.001$. ** $p < 0.01$. * $p < 0.05$. [a] 1 = initial schooling and 2 = other levels of schooling. [b] 1 = class coordinator and 2 = other roles. [c] 1 = Male and 2 = Female.

Finally, we used ANOVA and the independent samples *t*-test to test the second hypothesis. For this analysis, we re-coded the inclusive resources variable into a nominal variable with three categories based on the 25th ($P_{25}$ = 25) and 75th ($P_{75}$ = 34) percentiles (1—scores below 25, perception of low levels of inclusive resources; 2—scores between 26 and 33, perception of medium levels of inclusive resources; 3—scores above 34, perception of high levels of inclusive resources). Considering the overrepresentation of women in our sample, we conducted the independent *t*-test, using the Welch test (equal variances not assumed) due to the difference in the sample's number of men and women.

### 3. Results

The study aimed to investigate the relationship between inclusive practices and personal and career characteristics and to analyse differences in inclusive practices as functions of gender and career characteristics.

The sample mean suggested that teachers had high levels of perception of the implementation of inclusive practices (M = 66.49, SD = 28.94). However, the standard deviation was large, suggesting substantial variation in these perceptions.

We conducted multiple linear regression analysis (stepwise method) to assess the extent to which gender, inclusive resources, roles performed at school, and level of teaching predicted variance in inclusive practices.

The 'roles performed at school' variable was removed from the final model because it was not a significant predictor of inclusive practices. Table 3 presents the output of this final model. About 9.5% of the variance of inclusive practices was explained by inclusive resources (β = 0.265, $p < 0.001$), level of teaching (β = −0.160, $p < 0.001$), and gender (β = −0.094, $p < 0.01$) as statistically significant predictors.

**Table 3.** Multiple linear regression model (stepwise).

| Outcome | Predictors | R | Adj.$R^2$ | F | p | Beta Std. | p |
|---|---|---|---|---|---|---|---|
| Inclusive Practices | Inclusive resources | 0.313 | 0.095 | 33.33 | 0.001 | 0.265 | 0.001 |
| | Level of teaching | | | | | −0.160 | 0.001 |
| | Gender [a] | | | | | 0.094 | 0.003 |

[a] 1 = Male; 2 = Female.

Considering the results of the regression analyses, we sought to test our second hypothesis by performing a one-way analysis of variance to analyse the inclusive practices by career characteristics, namely, level of teaching (Table 4), inclusive resources (Table 5), and gender.

**Table 4.** Means, standard deviations, and one-way analyses of variance in level of teaching and inclusive practices.

| Level of Teaching | Inclusive Practices | | F | $\eta^2$ |
| --- | --- | --- | --- | --- |
| | *M* | *SD* | | |
| Early childhood | 67.03 | 5.87 | | |
| First level | 68.65 | 5.86 | | |
| Second level | 66.98 | 6.01 | | |
| Third level | 65.82 | 6.81 | 6.041 (6.923) *** | 0.038 |
| Secondary | 65.10 | 6.81 | | |
| More than one level | 64.78 | 8.38 | | |
| Special education | 67.74 | 6.94 | | |

*** $p < 0.001$.

**Table 5.** Means, standard deviations, and one-way analyses of variance in perception of inclusive resources and inclusive practices.

| Perception of Inclusive Resources | Inclusive Practices | | F | $\eta^2$ |
| --- | --- | --- | --- | --- |
| | *M* | *SD* | | |
| Perception of low levels of inclusive resources | 64.77 | 7.11 | | |
| Perception of medium levels of inclusive resources | 65.99 | 6.29 | 33.81 (2.923) *** | 0.068 |
| Perception of high levels of inclusive resources | 69.26 | 5.81 | | |

*** $p < 0.001$.

Teachers at the various levels of teaching were found to differ significantly in inclusive practices ($F(6, 923) = 6.041$, $p = 0.001$; $\eta^2 = 0.038$) (Table 4). Post hoc comparisons with Bonferroni correction were then performed to test differences between pairs of groups. These revealed that teachers at the first level of schooling reported more inclusive practices than did teachers in the third level of schooling, secondary education, and those teaching at more than one level of schooling.

Statistically significant differences were also found between teachers grouped according to their perception of inclusive resources ($F(2, 923) = 33.81$, $p = 0.001$; $\eta^2 = 0.068$) (Table 5). Post-hoc comparisons with Bonferroni correction indicated that teachers perceiving the most inclusive resources reported more inclusive practices.

Finally, we performed an independent samples *t*-test to assess whether male and female teachers differed in their inclusive practices. Women reported higher scores in inclusive practices (M = 66.89, SD = 6.47) than did men (M = 64.76, SD = 7.19), and this difference was statistically different ($t_{240.189} = 3.817$, $p < 0.001$, d = 0.322).

## 4. Discussion

The present study aimed to investigate the relationship between inclusive practices and teachers' personal and career characteristics and to analyse differences in inclusive practices as functions of gender and career-based characteristics. Inclusive education involves more than the placement of vulnerable students in a regular classroom and requires using inclusive teaching practices [13]. In this way, teachers assume a central role in inclusive education and consequently, investigating teacher-related factors can provide a better understanding of the use of inclusive practices. It is widely acknowledged that these practices enable learning and involvement, and adequately ensure that students' needs are met [11,18].

The first finding from this study was that teachers, on average, reported high levels of inclusive practice, although there was a high level of variability around this average. While these results present a positive picture of the state of inclusive practice in Portuguese schools, the observed high variability suggests the strong influence of context and other

variables. In this study, teachers' personal and career characteristics were our focus. From the regression analysis, we found that inclusive practice was significantly associated with the teachers' perceptions of inclusive resources, level of teaching, and gender. Teacher roles and years of experience were not significantly associated with inclusive practices, and so were excluded from the final model.

In this study, years of experience was not a significant predictor of inclusive practices, which contradicts research findings from Krischler et al. [32] and Schwab et al. [18]. This result must be interpreted considering the specificities of the Portuguese context because since the introduction of compulsory schooling, the conceptual framework for education remained practically unchanged until 2018 [33,37,45]. In addition, these findings corroborate other studies suggesting the influence of previous and intense contact with diversity on positive attitudes toward inclusive education and inclusive practices [46,47]. In this sense, the relationship between years of experience and inclusive practices may be mediated by several factors, as teaching experience by itself might not lead to more inclusive teaching practices. In this regard, the opportunity of contact with and experience in teaching students with different needs and who are more vulnerable to exclusion might improve knowledge and skills, providing different teaching practices and resources to work with diversity, and enable positive attitudes toward inclusion and inclusive practices [37,47].

Although the current literature presents a need to study the influence of teachers' roles in inclusion education [36], in this study, this variable was not significantly associated with inclusive practices. The timing of the data collection may be related to this result. Data collection was carried out soon after the enactment of Decree-Law 54/2018, which means that schools and professionals were involved in a process of reorganisation and changes in their formed roles. Another possible explanation for these findings is related to the way the variable was recoded. We recoded teachers' roles in a binary variable, considering the role of class coordinator vs. other roles, due to the proximity of the class coordinator to the students. Perhaps other roles make different contributions in the explanation of inclusive practices. Future studies can help to clarify this relation, namely, complementing quantitative studies with qualitative methodologies, to understand the typical activities relevant to the roles teachers embrace in the school context.

As expected, the perceived level of inclusive resources, level of teaching, and gender were all significantly associated with inclusive practices, accounting for 9.5% of the variance. This result corroborates other research findings and confirms the relevance of teachers' personal and career variables when analysing inclusive practices [11,13,15,20,31,34].

Among all the study's variables, teachers' perceptions of inclusive resources accounted for the largest portion of the variance in inclusive practices. Thus, inclusive practices can be said to be linked to inclusive resources. This result is consistent with previous research that showed teachers claim a need for additional resources in the classroom, especially specialized professionals, to support students who have an additional need of support [9,13,34]. This result also implies that the lack of resources seems to be a barrier to the implementation of the recent legal framework for inclusive education in Portugal. Interestingly, these results may also reflect teachers' perception of low levels of preparation and the need for more support to enhance self-efficacy toward inclusive education and to work in an inclusive way [19]. Moreover, the recent shift in the inclusive framework in Portugal now requires teachers to adopt a more proactive and collaborative approach, which might also contribute to the perception of the need for more support in the classroom to address the new challenges. Despite the relevance of resources for supporting inclusive practices, the teachers' perceptions may underline a need for more support for students with difficulties or additional support needs based on a deficit view of students' potential rather than a vision of effective inclusive practices for all students [13]. Hence, necessitating more than just resources in schools and classrooms, the use of inclusive practices requires the professional development of mainstream teachers and collaboration models for inclusion to facilitate the learning, participation, and involvement of all students. Indeed, support for colleagues enhances teachers' professional growth and learning [48]. As Vlachou et al. [49]

(2015, p. 562) stated, "well-informed, well-trained and sensitized teachers can create strong communities of practices that will enable them to more effectively demand and acquire the human and material resources they need, and deserve, to be able to respond to the divergent strengths and needs of all students".

Statistically significant differences in inclusive practices were found between teachers at different levels of education. Specifically, our results showed that teachers at lower levels of teaching used more inclusive practices than did those at the higher levels, which is consistent with previous research findings [11,20]. In Portugal, students in the first level of basic education typically have just one teacher who is responsible for most subjects, and this teacher usually follows their students across the first four school years. Starting at the second level of basic education, students then have one teacher per subject, with different teachers each school year. This organization in primary school allows for a closer relationship between teachers and students and enhances individualized practices and pedagogical differentiation.

We also found that male and female teachers differed in inclusive practices, with female teachers reporting greater use of inclusive practices than did male teachers. These results, which are similar to those of a study by Saloviita [26] on inclusive attitudes, may help to clarify the currently inconclusive literature. We note that this finding may be associated with the specific sample, as most participants were female (81.3%). However, this difference was studied by performing a Welch *t*-test, controlling the differences in the sample's proportions, and was broadly representative of the unequal distribution of gender in the education system in Portugal [41,42]. According to national statistics from 2019, 77.9% of teachers from early childhood to the secondary level were female [50].

Concerning the results of this study, it is appropriate to recognize several limitations. The first limitation relates to the reliance on teachers' self-reports to collect data. There are at least two problems associated with this type of measurement. A first problem is that when teachers recognize the relevance of inclusion in the education system, they may be biased to respond favourably rather than truthfully (social desirability bias). The second problem of self-rating is that it only represents teachers' perceptions. According to Sharma and Sokal [21], self-reporting measures assess behaviour and practice intentions more than the practices themselves. Therefore, in future research, it is important to incorporate other data collection methods, such as classroom observation, to add richness to self-reporting measures.

A second limitation of this study concerns the study's participants, because the results were based on teachers' perceptions. In future research, it will be useful to extend the current findings by exploring students' and other stakeholders' perceptions, to provide a deep understanding of classroom teaching practices [51]. Furthermore, we recognize the relevance of including students with different needs, as inclusive teaching practices might vary among student subgroups. For instance, Schwab et al. [18] reported that, in the same class, students perceived the inclusive practices of the same teacher differently. Hence, future studies may explore the variance within a class and student-related and school-related factors, as well as their influence on the use of inclusive practices.

In addition to those ideas already mentioned, we suggest that future research should consider qualitative methods to explore and understand dynamics associated with inclusive practices, as Laspina-Olmedo and Montero [52] suggested. Inclusive teaching practices seems to be a complex topic that requires multiple methods for a deep understanding of the factors that explain their use (or not).

Despite these limitations, this study represents an attempt to enhance the current understanding of teacher-related factors in the use of inclusive practices after enacting the recent legal framework on inclusive education in Portugal. Our findings highlight the need for additional resources, both human and physical, to support pedagogical practices. Digital resources are, after the recent COVID-19 pandemic, more available and friendly to students and teachers [53], so it would be helpful to understand how digital tools can support inclusive practices. Furthermore, these resources could be leveraged for students' learning

by providing pre-service and in-service teacher training to ensure that the teachers are able and sufficiently confident to use inclusive practices to address their students' diversity.

The results of the present study also invite a reflection on the relationship between teachers and students. The results regarding primary school teachers (with higher scores in inclusive practices) highlight a need to study differences between levels of teaching in the student–teacher relationship and their relationships with inclusive practices. In Portugal, several groups of schools are aiming for total autonomy to implement and evaluate educative frameworks. The study of teachers' practices in these schools, as well as the leadership practices and the differences among regular groups of schools, can add to existing knowledge and understanding of inclusive practices.

**Author Contributions:** Conceptualization, J.C. and H.A.; methodology, J.C., H.A., M.C. and H.F.; formal analysis, J.C. and H.A.; writing—original draft preparation, J.C. and H.A.; writing—review and editing, J.C., H.A., M.C. and H.F.; project coordination, M.C. All authors have read and agreed to the published version of the manuscript.

**Funding:** This research was funded by the Portuguese Foundation for Science (FCT) and Technology and the Portuguese Ministry of Science, Technology, and Higher Education through national funds within the framework of the Psychology of Development Research Centre—CIPD (grant number UIDB/04375/2020).

**Institutional Review Board Statement:** The study was conducted in accordance with the Declaration of Helsinki. Ethical review and approval were waived for this study due to the nonexistence of an Ethics Committee in the authors' institutions at the time of the research.

**Informed Consent Statement:** Informed consent was obtained from all subjects involved in the study. Written informed consent has been obtained to publish this paper.

**Data Availability Statement:** The raw data supporting the conclusions of this article will be made available by the authors, when requested.

**Conflicts of Interest:** The authors declare no conflict of interest.

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
