# Peer review of "From Policies to Practices: Factors Related to the Use of Inclusive Practices in Portugal"

_ejihpe, doi:10.3390/ejihpe13100158_

Round 1

Reviewer 1 Report

Dear Author/s,

Thank you so much for the opportunity to read this very interesting project.

Good Luck!

Kind Regards,

The Reviewer

Dear Author/s,

Professional proofreading is highly recommended.

Kind Regards,

The Reviewer

Author Response

Comments to the authors

Thank you for the possibility to read and discuss this interesting work dealing with policies and

practices of inclusive practices in Portugal. All recommendations and adjustments are suggestive to improve the paper`s quality from readers `perfectives.

Response: We thank the reviewer for this learning opportunity. We believe that this version of the manuscript is improved.

The following adjustment could be advantageous:

- Introduction section is too long and vague, and could split as Introduction and Literature

Review/Review of prior studies and clearly highlight why just for Portugal and NOT anywhere else;

Response: We made the changes suggested, and we believe that the introduction is now more clear and specific.

- All content under the current study title, could merge into literature review section;

Response: We agree with the author and merged the study presentation with the literature review

- In the data analysis section, it would be more clearer to the reader if the author/s would clearly highlight what is their response rate, the question fill up window and provide some evidence;

Response: We tried to clarify the procedures regarding the questionnaire administration, providing evidence from what happened in the study (lines 244-252). It was not possible to collect the response rate due to the fact that the questionnaire link was sent to the school directors, and they were responsible for delivering it to school teachers. 

- The research results would be more rigorous if the author could provide the evidence/reference of prior research results.

Response: Prior research results related to the use of the questionnaire only refer to the validation of the measure and it is available on the manuscript cited on measures section (lines 218-233). In the discussion, we corroborated the findings or discussed contradictory results by showing different studies and referenced them. For example, this can be seen in lines 359-360, as well as in lines 214-216 and 244-246.

 The following journal articles might add value:

Moura, A.; Fontes, F. Disabling experiences and inclusive school: reframing the debate in Portugal. Journal of Education Policy 2023, 1-17.

Alves, I. Enacting education policy reform in Portugal–the process of change and the role of teacher education for inclusion. European Journal of Teacher Education 2020, 43, 1, 64-82.

Laspina-Olmedo, T.; Montero, D. Competencia inclusiva en la práctica docente: análisis bibliográfico y propuesta de categorización. Alteridad 2023, 18, 2, 177-186.

Response: We thank the reviewer for the opportunity to analyze these new articles and we included them in the manuscript, enriching the document.

Reviewer 2 Report

Thank you for submitting your manuscript “From Policies to Practices: Factors Related to the Use of Inclusive Practices in Portugal” to the European Journal of Investigation in Health, Psychology, and Education. I am glad to be given this opportunity to read and review this manuscript. This manuscript explores the use of formal financial institutions and mobile money by persons with disabilities. I found this article very informative and crucial to expanding our understanding of access to financial services by persons with disabilities.

All sections of the manuscript are well-written with all important issues addressed (Abstract, Introduction, Materials and Methods, Result, and Discussion). The authors have reviewed relevant and current literature relating to their study. They have excellently explored the gaps in previous studies and clearly discussed how their study addresses the void in previous studies. All the concepts and variables have been clearly explained and well-adopted in the study.

Therefore, I recommend the publication of this manuscript in its present form.

Author Response

We would like to thank the reviewer for reading and reviewing this manuscript. 

Reviewer 3 Report

Thank you to the authors for the opportunity to read an interesting text . It deals with the important and actual issue of teachers' application of inclusive resources and practices.  The introduction recognizes the need for the study, arising from the assumptions of international law and the concordant changes taking place in the educational law of Portugal. The study was designed to examine the relationship between inclusive practices and the personal and professional characteristics of public and non-public school teachers, and to analyze differences in inclusive practices as a function of gender and professional characteristics. The research tools, and statistical analyses were correctly selected, and the procedure of sampling and data analysis was presented in detail. A significant research group participated in the study. Interesting results were obtained, useful for analyzing resources and shaping inclusive practices not only in Portugal. It may come as some surprise that years of experience was not a significant predictor of inclusive practices. This result was skillfully justified in the discussion with reference to the Portuguese context, although it may also be of interest in the context of international comparisons . The importance of resources and the use of inclusive practices for the success of inclusive eduaction was pointed out. The selection of sources is correct and representative. 

Author Response

(The authors gave the same response as above.)
